# Emergence failure of early epidemics: A mathematical modeling approach

**Romulus Breban**[ID]*

Institut Pasteur, Unité d'Epidémiologie des Maladies Emergentes, Paris, France

* Romulus.Breban@pasteur.fr

**Data Availability Statement:** The manuscript does not contain nor is based on raw data. The minimal data set and any additional data required to replicate our study findings in their entirety are included in the main text of the manuscript.

## Abstract

Epidemic or pathogen emergence is the phenomenon by which a poorly transmissible pathogen finds its evolutionary pathway to become a mutant that can cause an epidemic. Many mathematical models of pathogen emergence rely on branching processes. Here, we discuss pathogen emergence using Markov chains, for a more tractable analysis, generalizing previous work by Kendall and Bartlett about disease invasion. We discuss the probability of emergence failure for early epidemics, when the number of infected individuals is small and the number of the susceptible individuals is virtually unlimited. Our formalism addresses both directly transmitted and vector-borne diseases, in the cases where the original pathogen is 1) one step-mutation away from the epidemic strain, and 2) undergoing a long chain of neutral mutations that do not change the epidemiology. We obtain analytic results for the probabilities of emergence failure and two features transcending the transmission mechanism. First, the reproduction number of the original pathogen is determinant for the probability of pathogen emergence, more important than the mutation rate or the transmissibility of the emerged pathogen. Second, the probability of mutation within infected individuals must be sufficiently high for the pathogen undergoing neutral mutations to start an epidemic, the mutation threshold depending again on the basic reproduction number of the original pathogen. Finally, we discuss the parameterization of models of pathogen emergence, using SARS-CoV1 as an example of zoonotic emergence and HIV as an example for the emergence of drug resistance. We also discuss assumptions of our models and implications for epidemiology.

## Introduction

Emerging infectious diseases (EIDs) are explained by the World Health Organization (WHO) as the diseases *whose incidence in humans has increased during the last two decades or which threaten to increase in the near future. The term includes newly-appearing infectious diseases or those spreading to new geographical areas. It also refers to those that were easily controlled by chemotherapy and antibiotics but have developed antimicrobial resistance* [1]. New human pathogens causing EIDs are continuously being discovered [2, 3], whilst the frequency of EIDs

**Funding:** The author received no specific funding for this work.

**Competing interests:** The author has declared that no competing interests exist.

outbreaks has steadily increased [4]. Therefore, EIDs pose an ever-increasing threat [5, 6]. Of equal concern are zoonoses, which could result in new human pathogens [7–9] and thus EIDs.

Chances are that EIDs initiate new epidemics, leading, in the long term, to endemic disease. The typical abstraction is that an epidemic starts with a *patient zero* [10, 11], in a disease-naïve community, who is expected to further pass the infection to $R_0$ other individuals; n.b., $R_0 > 1$. Then, each newly infected individual is expected to further pass the infection to $R_0$ other individuals; therefore, cases are expected to increase exponentially, at least initially. Two possibilities have been identified for how a patient zero might occur.

First, a patient zero may occur as a case of a new disease or an imported case. Over the past two decades, the WHO has repeatedly warned against importation of cases of polio, MERS, Zika, etc.: *A single imported case can reignite an outbreak or bring cases to a new area, if preparedness measures are weak* [12]. Furthermore, the traffic of animals [13] has contributed to the importation of zoonotic diseases [14–16]. Traffic of passengers, live animals and merchandise, altogether, has also contributed to the dispersal of disease vectors (e.g., mosquitos, ticks, etc.), which can import disease from endemic areas [17, 18]. These epidemiological setups are known under the name of *disease invasion*. The mathematical foundations to conceptualize disease invasion were laid by Kendall [19] and Bartlett [20] more than 50 years ago. More recently, Allen et al. [21, 22] proposed generalizations of the mathematical formalism describing disease invasion. Chowell et al. [23] review mathematical models of early epidemics.

Secondly, a patient zero can occur through pathogen evolution, from an individual infected with a disease that is poorly transmitted, so the basic reproduction number is $R_0' < 1$; we call this individual *patient minus one*. Evidently, patient minus one cannot trigger a mass-scale epidemic, but, through pathogen evolution, he may become a patient zero who can. This phenomenon can arise as the pathogen mutates and adapts within the human host. Thus, the initial pathogen population infecting patient minus one can be replaced by a mutant pathogen population such that the patient minus one becomes a patient zero, transmitting the mutant pathogen with $R_0 > 1$. Antia et al. [24] called this phenomenon *pathogen emergence*, studied it numerically, and used it to illustrate the emergence of SARS-CoV1 disease and HIV disease from wild-type pathogens. Subsequently, pathogen emergence has been thoroughly studied.

Upon analysis of the key biological principles [25–32], an array of factors has been empirically identified to justify why some pathogens do emerge while others do not. These include the pathogen's host range [2, 33–37], host susceptibility [2, 38, 39], host-genetic diversity [40–42] and species richness [43, 44], contact patterns in the host population [2, 38, 45–49], mechanisms of pathogen adaptation [38, 50], pathogen taxonomic classification [33, 34], pathogen generation time [33, 34] and growth rate [46], pathogen mutation dynamics [33, 34, 38, 46, 51], and environmental changes [5, 49, 52, 53].

The numerical framework by Antia et al. [24], based on branching processes, was later amended to study the impact of biological factors on various aspects of emergence, such as host-type heterogeneity [54, 55], adaptation pathways [55, 56], spatial heterogeneity [57], ongoing reservoir interactions [58], host–population viscosity [59], and surveillance conditions [60]. Furthermore, complex networks [56, 61, 62] have been employed to study host–contact patterns that may favor EIDs epidemics, modeled as bond percolations within complex networks. The emergence of vector-borne pathogens [63–65] and parasitic zoonoses [66, 67] have been given considerably less attention.

The choice of mathematical/numerical framework is very important to solve emergence problems. Kendall [19] and Bartlett [20] used Markov chains to solve analytically for the probability that an imported case is a patient zero, who initiates an epidemic, or else the transmission process he initiates goes extinct. Their results would have been very different had they

used branching processes instead of Markov chains. We say that the extinction of a Markov chain occurs in time; i.e., after an arbitrarily large time interval, the process is extinct. In contrast, the extinction of a branching process occurs in the generation number; after an arbitrarily large number of generations, all subsequent generations have zero individuals. The relationship between time and generation number is complex; individuals ordered by time of infection may not also be ordered by generation number. For a brief illustration, we invoke the Crump–Mode–Jagger continuous-time branching process, corresponding to the *SI* model of disease invasion [68–70]. This model has a Markov chain embedding the so-called *Kendall process*, whose extinction probability is $1/R_0$ [19]. However, the model also has an embedding consisting of a Bienaymé–Galton–Watson discrete-time branching process, where the distribution of secondary cases is Poisson. The extinction probability $p$ of the branching process verifies $p = \exp[-R_0(1 - p)]$. It is obvious that $p = 1/R_0$ is not a solution; therefore, the probability of extinction of the discrete-time branching process differs from that of the Markov chain.

In this paper, we generalize the work of Kendall [19] and Bartlett [20] about disease invasion, to include the mathematical principles of pathogen emergence. Our primary focus is the analytic derivation of the probability that a patient minus one fails to initiate an epidemic of a directly transmitted or vector-borne disease. This probability could be retrieved from data structured as time series of cases resulting from outbreak investigations. Data on the generation of disease transmission requires contact tracing and is more costly to obtain. We discuss our results not only in the language of zoonotic emergence but also in the language of emergence of drug-resistant pathogens, such as HIV.

## Pathogen emergence in the SIR formalism

The flow diagram of our model is illustrated in Fig 1. We express the population-level model using ordinary differential equations (ODEs), to set up the context of zoonotic emergence from a wild-type strain

$$\frac{dS}{dt} = \pi - S\left(\beta'\frac{I'}{N} + \beta\frac{I}{N}\right) - \mu S, \tag{1}$$

$$\frac{dI'}{dt} = S\left(\beta'\frac{I'}{N}\right) - (\mu + \gamma')I' - m'I', \tag{2}$$

$$\frac{dI}{dt} = m'I' + S\left(\beta\frac{I}{N}\right) - (\mu + \gamma)I, \tag{3}$$

$$\frac{dR'}{dt} = \gamma'I' - \mu R', \tag{4}$$

$$\frac{dR}{dt} = \gamma I - \mu R. \tag{5}$$

Here, $S$ represents the number of susceptible individuals. $I'$ and $I$ represent the numbers of infectious individuals with the wild-type and mutant strains, respectively. $R'$ and $R$ represent the numbers of individuals recovered from infections with the wild-type and mutant strains, respectively. The symbol $N$ stands for the total population size; i.e., $N = S + I' + I + R' + R$. The parameters $\pi$ and $\mu$ are demographic and designate, respectively, the inflow of susceptibles and per-capita, disease-unrelated, mortality rate. The parameters $\beta'$ and $\gamma'$ characterize the transmissibility and per-capita recovery rate for the disease caused by the wild type-strain; $\beta$ and $\gamma$

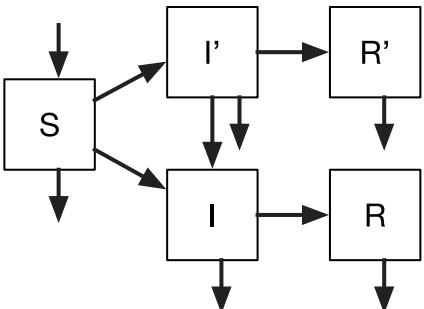

**Fig 1. Flow diagram of our proposed model of zoonotic emergence from a wild-type strain, which is directly transmitted.** The model has SIR structure, where the primed variables relate to the infection with the zoonotic strain. The arrow from $I'$ to $I$ indicates the process of mutation at the population level that turns a poorly transmissible zoonotic strain into a pandemic strain.

are similar, for the mutant strain. The symbol $m'$ stands for the rate at which patient minus one becomes a patient zero, the so-called *mutation rate at the population level*. It is assumed that, due to their genetic proximity, the wild type and the mutant strains can be considered in relationship of perfect cross immunity. According to the next-generation method [71], the basic reproduction number of the above model is the maximum of $\mathbb{R}'_0 = \beta'/(\gamma' + m' + \mu)$ and $\mathbb{R}_0 = \beta/(\gamma + \mu)$; n.b., our problem setup assumes $\mathbb{R}'_0 < 1$ and $\mathbb{R}_0 > 1$, so the overall reproduction number of the model is $\mathbb{R}_0$.

We consider Eqs (1)–(5) at *disease invasion*, where $I'$, $I$, $R'$ and $R$ are small (i.e., $S/N \approx 1$), and the cross-immunity assumption is not strictly needed, because the number of cross infections is negligible. We investigate time scales much less than individual's expected lifetime and neglect $\mu$ next to $\gamma$. This represents a singular perturbation for the model (1)–(5), but applies well to the model of disease invasion, where we have $\mathbb{R}'_0 \approx \beta'/(\gamma' + m')$ and $\mathbb{R}_0 \approx \beta/\gamma$, and maintain the requirements $\mathbb{R}'_0 < 1$ and $\mathbb{R}_0 > 1$. The model of disease invasion is expressed by two linear ODEs

$$\frac{dI'}{dt} = \beta'I' - \gamma'I' - m'I', \tag{6}$$

$$\frac{dI}{dt} = \beta I + m'I' - \gamma I. \tag{7}$$

Hence, the model cannot address problems of population extinction. Solutions for the population numbers, with positive initial conditions, are sums of exponentials, and do not reach zero after an arbitrarily large time interval. To address this shortcoming, we introduce a continuous-time Markov chain, with two nonnegative-integer random variables, $i'(t)$ and $i(t)$, such that the expectation values of $i'(t)$ and $i(t)$ satisfy Eqs (6) and (7) (i.e., $\langle i'(t) \rangle = I'(t)$ and $\langle i(t) \rangle = I(t)$). This adds realism to the ODE model of disease invasion and opens the discussion about epidemic extinction. The point processes of the Markov chain and their corresponding rates are listed in Table 1. Indeed, straightforward calculations, using moment closure techniques [72–75], show that the expectation values of $i'(t)$ and $i(t)$ satisfy Eqs (6) and (7). The moment expansion closes exactly at the expectation because all the rates of the Markov chain are linear in the stochastic variables. Furthermore, we note that if $i'$ vanishes, then the Markov chain reduces to the Kendall process [19].

Markov chains can be naturally used to model population extinction. For the Markov chain defined in Table 1, the time to extinction is given by $\inf\{t \geq 0 : i'(t) = 0, i(\mathrm{I}t) = 0\}$, where $i(t)$ and $i'(t)$ depend on the model parameters and initial condition. The extinction probability is

**Table 1. Markov chain for the model with direct transmission: Stochastic processes and their corresponding rates.**

| Count $j$ | Process | Definition | Rate $\mathcal{R}_j$ |
|---|---|---|---|
| 1 | Infection with the wild-type strain | $i' \to i' + 1$ | $\beta' i'$ |
| 2 | Recovery from infection with the wild-type strain | $i' \to i' - 1$ | $\gamma' i'$ |
| 3 | Mutation of the wild-type strain | $i' \to i' - 1, i \to i + 1$ | $m' i'$ |
| 4 | Infection with the mutant strain | $i \to i + 1$ | $\beta i$ |
| 5 | Recovery from infection with the mutant strain | $i \to i - 1$ | $\gamma i$ |

the fraction of times the Markov chain trajectories go extinct. However, to obtain the probability that patient minus one fails to initiate an epidemic, we do not invoke the definition. Rather, we invoke the *Gillespie's exact method* [76], an algorithm to integrate continuous-time Markov chains, where each run of the algorithm provides a realization of the stochastic model. Gillespie's algorithm is a repetition of two basic steps. First, using the initial conditions and the model parameters (see Table 1), the total rate $\mathcal{R} = \sum_{j=1}^{5} \mathcal{R}_j$ is calculated, and the time to the next process is chosen as an exponential deviate with the total rate $\mathcal{R}$; i.e., $\Delta t \sim \mathrm{Exp}(\mathcal{R})$. Secondly, the Process $j$ is chosen to be performed next with probability $\mathcal{R}_j / \mathcal{R}$, and the population variables are updated accordingly. The algorithm repeats from step one, using the current population variables.

Using Gillespie's algorithm, with patient minus one as initial condition (i.e., $i'(0) = 1$, $i(0) = 0$), we obtain the probabilities that the individual participates in each available process after one time step. Because $i(0) = 0$, only Processes 1–3 contribute non-zero rates; their corresponding probabilities to occur as the first process are $\beta'/(\beta' + \gamma' + m')$, $\gamma'/(\beta' + \gamma' + m')$, and $m'/(\beta' + \gamma' + m')$. We denote by $p'_1$ the probability that the Markov chain, starting with patient minus one, goes extinct and pathogen emergence fails. We rationalize why the Markov chain went extinct.

Extinction occurred because, at the first process: 1) patient minus one infected another individual with probability $\beta'/(\beta' + \gamma' + m')$, and pathogen emergence from both individuals failed with total probability $p'^2_1$, or 2) patient minus one recovered, with probability $\gamma'/(\beta' + \gamma' + m')$, or 3) became a patient zero with probability $m'/(\beta' + \gamma' + m')$ and the epidemic starting with patient zero went extinct with probability $p_1$. For the probability of stochastic extinction of the SIR model at disease invasion (i.e., the ODE version is Eq (7) with $m' = 0$ and $I' = 0$), Kendall [19] obtained $p_1 = 1/\mathrm{R}_0$ if $\mathrm{R}_0 = \beta/\gamma > 1$ and $p_1 = 1$, otherwise. Hence, using the above-mentioned results, we write a self-consistent equation for $p'_1$,

$$p'_1 = \frac{p'^2_1 \, \beta'}{\beta' + \gamma' + m'} + \frac{\gamma'}{\beta' + \gamma' + m'} + \frac{p_1 \, m'}{\beta' + \gamma' + m'}, \tag{8}$$

whose solution is expressed as a function of $p_1$ and just two ratios of the parameters $\beta'$, $\gamma'$ and $m'$. For these, we chose $\mathrm{R}'_0 = \beta'/(\gamma' + m')$ and $\mathrm{m}' \equiv m'/(\gamma' + m')$, where $\mathrm{m}'$ is interpreted as the expected number of opportunities patient minus one has to become patient zero, during his entire infectious period. Eq (8) is quadratic in $p'_1$ and has a unique subunitary solution, which can be written as

$$p'_1(\mathrm{R}'_0, \mathrm{m}'; p_1) = \frac{1 + \mathrm{R}'^{-1}_0}{2} \left[ 1 - \sqrt{1 - \frac{1 - \mathrm{m}'(1 - p_1)}{\mathrm{R}'_0 \left( \frac{1 + \mathrm{R}'^{-1}_0}{2} \right)^2}} \right] \equiv F'_{\boldsymbol{\mu}}(p_1), \tag{9}$$

where we introduce the function $F'_{\boldsymbol{\mu}}(\cdot)$ depending on a set of two parameters $\boldsymbol{\mu} \equiv (\mathrm{R}'_0, \mathrm{m}')$.

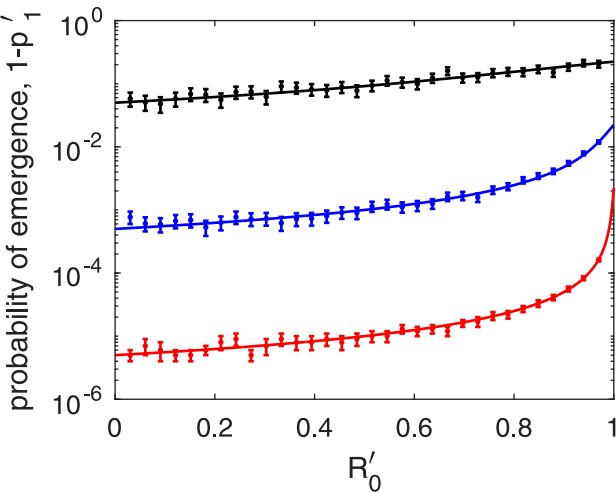

**Fig 2. The probability of epidemic emergence starting from patient minus one,** $1 - p'_1$**, as a function of** $R'_0$**, the basic reproduction number of the infection associated to patient minus one.** We assume that the pathogen is directly transmitted and choose three values for the mutation parameter: $m' = 0.1$ (black plot), $m' = 0.001$ (blue plot) and $m' = 0.00001$ (red plot). We also assume that the basic reproduction number of the emerged pathogen is $R_0 = 2$. The curves are obtained using Eq (9), and the points with error bars are obtained through brute-force integration of the Markov chain described in Table 1. Note that, with decreasing mutation rate, the probability curve develops a hyperbolic trend in the vicinity of $R'_0 = 1$.

In Fig 2, we compare the output of Eq (9) (continuous curves) with results from the brute-force integration of the Markov chain described in Table 1 (points with error bars). We integrated the Markov chain repeatedly, using the Gillespie algorithm and starting with one patient minus one as initial condition; i.e., $i'(0) = 1$, $i(0) = 0$. We stopped the integration at time $t$ if $i'(t) + i(t) = 0$ or $i(t) > 10,000$, under the assumption that once $i(t)$ reaches 10,000, further extinction is highly improbable. We used the fraction of times the population went extinct as an estimate of $p'_1$. Furthermore, for the error bars, we calculated the 95% confidence interval of each probability of extinction estimate using binomial statistics.

The black plot refers to $m' = 0.1$, the blue plot to $m' = 0.001$, and the red plot to $m' = 0.00001$. The horizontal axis represents $R'_0$, which goes from 0 to 1. We choose $R_0 = 2$. The Markov-chain parameterization compatible with this choice of $m'$, $R'_0$ and $R_0$ is not unique. We choose the following parameters: $\beta' = 1$, $m = \beta' m'/R'_0$, $\gamma' = \beta'/R'_0 - m'$, $\beta = 1$ and $\gamma = 0.5$. The numbers of repeat integrations are, respectively 1000, 100,000, and 10,000,000, such that error bars are clearly discernible in the linear-logarithmic scale. Overall, there is good agreement between the colored lines and the corresponding points with error bars. Furthermore, increasing the number of repeat integrations by a factor of 10 (results not shown), the error bars decrease considerably in height and the agreement is much improved.

Note that, with decreasing mutation rate, the probability curve develops a hyperbolic trend in the vicinity of $R'_0 = 1$. This is confirmed by analytic results. Assuming that the mutation rate of the pathogen is low, $p'_1$ depends weakly on the variable $m'$, and the first-order series expansion in $m'$ yields

$$p'_1(R'_0, m'; p_1) \approx 1 - \left[\frac{m'(1 - p_1)}{1 - R'_0}\right]. \qquad (10)$$

Hence, as $R'_0$ approaches 1 from below, $R'_0$ becomes the most important parameter determining changes in the probability of emergence. Previous numerical results [24, 57, 60] support a

similar parameter classification. It is important to note that Eq (10) is just an asymptotic form; the singularity in the denominator does not occur in Eq (9). Finally, note that $R_0' = 0$ yields a simple formula

$$p_1'(0, m'; p_1) = 1 - m'(1 - p_1),\qquad(11)$$

which can be read as follows. The probability of emergence $(1 - p_1')$ equals the probability of acquiring the mutant strain $m'$ times the probability that patient zero triggers an epidemic $(1 - p_1)$.

## The case of *n*-step mutation

A meaningful and straightforward generalization of the above problem is where the relevant mutation occurs in a sequence of *n* steps [24, 57, 60]. Tracking the dynamics of all the pathogen-subpopulations involved in the mutation sequence, the single mutation model (6) and (7) becomes

$$\frac{dI^{(n)}}{dt} = \beta^{(n)}I^{(n)} - \gamma^{(n)}I^{(n)} - m^{(n)}I^{(n)},\qquad(12)$$

$$\frac{dI^{(n-1)}}{dt} = \beta^{(n-1)}I^{(n-1)} - \gamma^{(n-1)}I^{(n-1)} + m^{(n)}I^{(n)} - m^{(n-1)}I^{(n-1)},\qquad(13)$$

$$\vdots$$

$$\frac{dI}{dt} = \beta I - \gamma I + m'I'.\qquad(14)$$

We assume that the initial, wild-type pathogen, with $R_0^{(n)} \equiv \beta^{(n)}/(\gamma^{(n)} + m^{(n)}) < 1$, undergoes *n* successive step-mutations. The first *n* − 1 mutated pathogens have $R_0^{(k)} \equiv \beta^{(k)}/(\gamma^{(k)} + m^{(k)}) < 1$; *k* = *n* − 1, *n* − 2, . . ., 1. Lastly, the *n*th mutation yields a stable pathogen; i.e., $m^{(0)} \equiv m = 0$, $\beta^{(0)} \equiv \beta$, $\gamma^{(0)} \equiv \gamma$ and $R_0^{(0)} \equiv R_0 = \beta/\gamma > 1$. Furthermore, the Markov chain described in Table 1 can be straightforwardly generalized to include all the populations infected with mutant pathogens described by Eqs (12)–(14). The equations for the extinction probability of the generalized Markov chain, starting with patient minus *n*, $p_1^{(n)}$, are

$$p_1^{(n)} = \frac{(p_1^{(n)})^2\,\beta^{(n)}}{\beta^{(n)} + \gamma^{(n)} + m^{(n)}} + \frac{\gamma^{(n)}}{\beta^{(n)} + \gamma^{(n)} + m^{(n)}} + \frac{p_1^{(n-1)}\,m^{(n)}}{\beta^{(n)} + \gamma^{(n)} + m^{(n)}},\qquad(15)$$

$$\begin{aligned}
p_1^{(n-1)} &= \frac{(p_1^{(n-1)})^2\,\beta^{(n-1)}}{\beta^{(n-1)} + \gamma^{(n-1)} + m^{(n-1)}} + \frac{\gamma^{(n-1)}}{\beta^{(n-1)} + \gamma^{(n-1)} + m^{(n-1)}} \\
&\quad + \frac{p_1^{(n-2)}\,m^{(n-1)}}{\beta^{(n-1)} + \gamma^{(n-1)} + m^{(n-1)}},
\end{aligned}\qquad(16)$$

$$\vdots$$

$$p_1 = \frac{p_1^2\,\beta}{\beta + \gamma} + \frac{\gamma}{\beta + \gamma}.\qquad(17)$$

Equation $(n - k + 1)$ in the above system of $(n + 1)$ equations is quadratic in $p_1^{(k)}$ and can be solved analytically. Assuming the parameterization provided by $R_0^{(k)} = \beta^{(k)}/(m^{(k)} + \gamma^{(k)})$ and $m^{(k)} \equiv m^{(k)}/(m^{(k)} + \gamma^{(k)})$, we denote the solution for $p_1^{(k)}$ as a function $F_{\boldsymbol{\mu}}^{(k)}(p_1^{(k-1)})$ depending only on the parameter set $\boldsymbol{\mu}^{(k)} \equiv (R_0^{(k)}, m_h^{(k)})$. The solution for the probability $p_1^{(n)}$ is then formally obtained through $n$ function compositions

$$p_1^{(n)} = F_{\boldsymbol{\mu}}^{(n)}(F_{\boldsymbol{\mu}}^{(n-1)}(... F_{\boldsymbol{\mu}}'(p_1)...)), \qquad (18)$$

where $p_1 = 1/R_0 = \gamma/\beta$.

Eq (18) has many parameters and potential to describe a variety of biological setups. We formalize a simple problem, with reduced number of parameters, that may be relevant for further qualitative understanding of the evolutionary path to pathogen emergence. We consider the case where, once imported within host, the wild-type pathogen (i.e., strain $n$) finds itself in an unstable evolutionary equilibrium. Most mutations are neutral, not changing $R_0^{(k)}$ or $m^{(k)}$, and non-neutral mutations occur very rarely. We investigate the probability that pathogen emergence fails during an arbitrary long chain of neutral mutations. For this, we interpret Eq (18) as a backward iteration of $n$ steps along the neutral mutation chain, where pathogen parameters stay the same: $R_0^{(k)} = R_0' = \beta'/(\gamma' + m') < 1$ and $m^{(k)} = m' < 1$, $\forall k = 1, \ldots, n$. We write a one-dimensional, discrete dynamical system or map [77] for the probability of extinction in the unit interval [0, 1]

$$p_1^{(k+1)} = F_{\boldsymbol{\mu}}'(p_1^{(k)}), \qquad (19)$$

where the explicit form of $F_{\boldsymbol{\mu}}'(\cdot)$ can be read from Eq (9).

Going backwards in the mutation chain, the unstable equilibrium appears as a stable equilibrium or fixed point. The map $F_{\boldsymbol{\mu}}'(\cdot)$ has two fixed points. First, $\hat{p}_1 = 1$ is a fixed point for all the parameter space, with the stability condition

$$\left| \frac{\partial F_{\boldsymbol{\mu}}'}{\partial p_1} \right|_{\hat{p}_1 = 1} = \frac{m'}{1 - R_0'} < 1 \Leftrightarrow \frac{\beta'}{\gamma'} < 1. \qquad (20)$$

If $\hat{p}_1$ is stable, then orbits will be attracted to $\hat{p}_1$ and pathogen extinction is guaranteed; i.e., with probability $\hat{p}_1 = 1$. If $\hat{p}_1$ is unstable, then orbits will wander away from $\hat{p}_1$, pathogen extinction is not guaranteed, and the population of pathogens undergoing neutral drift may persist.

The second fixed point of $F_{\boldsymbol{\mu}}'(\cdot)$ is $\tilde{p}_1 = (1 - m')/R_0'$, and acquires biological and probabilistic interpretation as the first fixed point, $\hat{p}_1$, becomes unstable; i.e.,

$$\tilde{p}_1 = (1 - m')/R_0' < 1 \Leftrightarrow \frac{\beta'}{\gamma'} > 1. \qquad (21)$$

The biological interpretation is that each strain in the mutation chain, except the stable one, has $R_0' < 1$. They are guaranteed to go extinct. However, if they replicate and mutate fast enough [i.e., with mutation rate $m' > (1 - R_0')$], then they have the chance $(1 - \tilde{p}_1)$ to advance along the mutation chain before going extinct. The neutral-mutation dynamic can persist (i.e.,

extinction does not occur with probability 1) if and only if

$$\left| \frac{dF'_{\boldsymbol{\mu}}}{dp_1} \right|_{\tilde{p}_1 = \gamma'/\beta'} = \frac{\mathrm{m}'}{2\mathrm{m}' + \mathrm{R}'_0 - 1} < 1 \Leftrightarrow \frac{\beta'}{\gamma'} > 1, \tag{22}$$

which is satisfied owing to [Eq (21)].

This fixed-point structure is reminiscent of a transcritical bifurcation. Indeed, [Fig 3] shows two graphs of $F'_{\boldsymbol{\mu}}(p_1^{(k)})$, for two different parameter sets, suggesting that $F'_{\boldsymbol{\mu}}(p_1^{(k)})$ undergoes a transcritical bifurcation. By the transcritical bifurcation theorem [78], a map $G_{\mu}(\cdot)$, depending on one variable and having one parameter $\mu$, undergoes a transcritical bifurcation at $(\hat{p}_1^*, \mu^*)$, if and only if

$$G_{\mu}(\hat{p}_1^*) = \hat{p}_1^*, \tag{23}$$

$$\lambda(\mu^*) = 1, \;\; \text{where} \;\; \lambda(\mu) \equiv \left. \frac{\partial G_{\mu}(p_1^{(k)})}{\partial p_1^{(k)}} \right|_{p_1^{(k)} = p_1^*}, \tag{24}$$

$$\left. \frac{d\lambda(\mu)}{d\mu} \right|_{\mu = \mu^*} \neq 0, \;\;\; \left. \frac{\partial^2 G_{\mu^*}(p_1^{(k)})}{\partial^2 p_1^{(k)}} \right|_{p_1^{(k)} = \hat{p}_1^*} \neq 0. \tag{25}$$

Redefining the parameter set

$$\boldsymbol{\mu}' = (\mathrm{R}'_0, \mathrm{m}'_h) \rightarrow \left( \mathrm{R}'_0, \lambda \equiv \frac{\mathrm{m}'_h}{1 - \mathrm{R}'_0} \right) \tag{26}$$

and choosing $\lambda$ for the bifurcation parameter, the map $F'_{\boldsymbol{\mu}}(\cdot) \equiv F'_{\lambda}(\cdot)$ satisfies the conditions

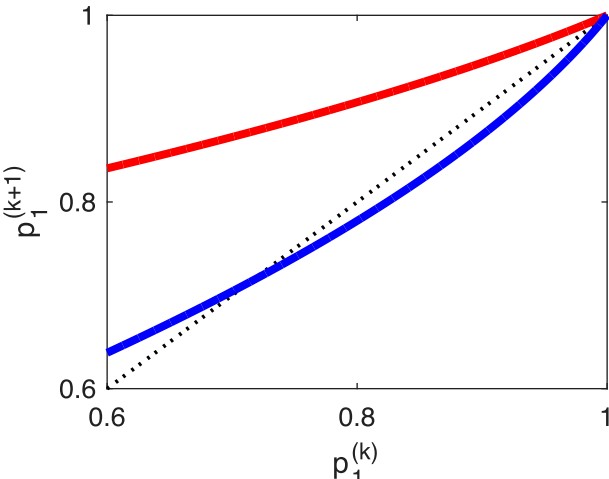

**Fig 3. Transcritical bifurcation of the neutral-genetic dynamics for the pathogen with direct transmission mechanism.** We illustrate $p_1^{(k+1)}$ versus $p_1^{(k)}$ or $F'_{\boldsymbol{\mu}}(\cdot)$. The parameter sets are $(R'_0, \mathrm{m}') = (0.7, 0.17)$ for the red curve, and $(R'_0, \mathrm{m}') = (0.7, 0.5)$ for the blue curve. The dotted line represents the first bisect line. It appears that $p_1^{(k+1)}(\cdot)$ undergoes a transcritical bifurcation at $(p_1^{(k)}, p_1^{(k+1)}) = (1, 1)$. That is, with the variation of the second parameter, $\mathrm{m}'$, the red curve becomes the blue curve, where $(1, 1)$ always remains a fixed point. Note that the red curve has a single stable point at $(1, 1)$, whereas the blue curve has this point unstable and the interior fixed point stable. This is a consequence of the fact that we restricted the domain of $p_1^{(k)}$ to $[0, 1]$, as it represents a probability. If we extend the domain of $p_1^{(k)}$ to the whole real axis, then each curve has one stable and one unstable fixed point.

(23) and (24) for having a transcritical bifurcation at $(\hat{p}_1^*, \lambda^*) = (1, 1)$; the conditions (25) are verified numerically.

This setup can offer qualitative understanding for the population dynamics of diseases where each pathogen strain undergoes frequent neutral mutations, has $R_0' < 1$, and necessarily goes extinct within each patient. However, the entire mixed-pathogen population can persist and be further transmitted to other individuals, with a basic reproduction number $\tilde{R}_0 > 1$. Using the results of Kendall [19], the second fixed point $\tilde{p}_1$ can be formally interpreted as the probability of epidemic extinction for a fictitious pathogen (i.e., all pathogen strains together considered as a single strain) with $\tilde{R}_0 = 1/\tilde{p}_1 > 1$, which does not undergo neutral mutation.

## Pathogen emergence of vector-borne diseases

Understanding epidemic extinction and emergence for diseases like Chikungunya, dengue and malaria requires mathematical models describing vector-borne transmission. Among others, Bartlett [20], Griffiths [79], and Lloyd [65] addressed epidemic extinction using stochastic models of vector-borne disease invasion. The probability of extinction was obtained analytically, as a function of the model parameters, where transmission starts from any number of patients zero and vectors zero. Notably, if transmission starts from one patient zero and one vector zero, then the probability of epidemic extinction is $1/R_0$ [20, 79]. In contrast to the $R_0$ recipe by van den Driessche and Watmough [71], here we say that $R_0$ describes the vector-borne transmission from host to host, just like direct transmission, ignoring the details of transmission from host to vector or vector to host. Hence, our $R_0$ is the square of that proposed by van den Driessche and Watmough [71].

The spread of emergent vector-borne diseases has been studied using deterministic models alone. The most popular problem has been, by a large margin, the epidemiology of malarial drug resistance [80, 81]. Here, as well, we start with an ODE framework to set up the epidemiological context. We propose an SI/SIR structure for the vector/host population compartments, where the variables and parameters carry, respectively, the subscripts $v$ and $h$. The flow diagram is illustrated in Fig 4.

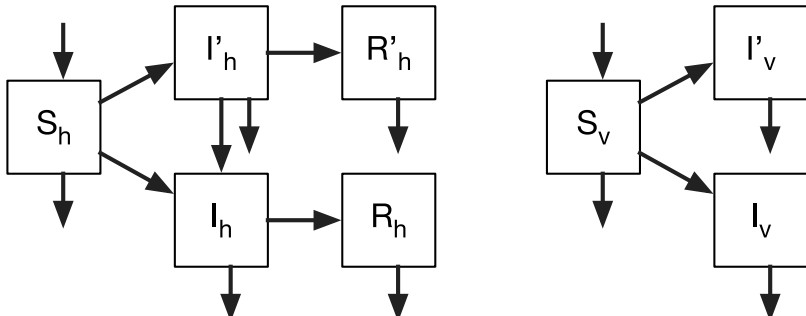

**Fig 4. Flow diagram of our model of zoonotic emergence from a wild-type strain, which is vector-borne.** The model has SIR structure for the human population and SI structure for the vector population. The primed variables relate to the infection with the zoonotic strain. The arrow from $I_h'$ to $I_h$ indicates the process of mutation in the human population that turns a poorly transmissible zoonotic strain into a pandemic strain.

Following previous studies, we assume that pathogen mutation occurs only within hosts, and vectors do not recover from infection.

$$\frac{dS_h}{dt} = \pi_h - S_h\left(\beta'_v \frac{I'_v}{N_v} + \beta_v \frac{I_v}{N_v}\right) - \mu_h S_h, \tag{27}$$

$$\frac{dI'_h}{dt} = S_h\left(\beta'_v \frac{I'_v}{N_v}\right) - (\mu_h + \gamma'_h)I'_h - m'_h I'_h, \tag{28}$$

$$\frac{dI_h}{dt} = m'_h I'_h + S_h\left(\beta_v \frac{I_v}{N_v}\right) - (\mu_h + \gamma_h)I_h, \tag{29}$$

$$\frac{dR'_h}{dt} = \gamma'_h I'_h - \mu_h R'_h, \tag{30}$$

$$\frac{dR_h}{dt} = \gamma_h I_h - \mu_h R_h, \tag{31}$$

$$\frac{dS_v}{dt} = \pi_v - S_v\left(\beta'_h \frac{I'_h}{N_h} + \beta_h \frac{I_h}{N_h}\right) - \mu_v S_v, \tag{32}$$

$$\frac{dI'_v}{dt} = S_v\left(\beta'_h \frac{I'_h}{N_h}\right) - (\mu_v + \gamma'_v)I'_v, \tag{33}$$

$$\frac{dI_v}{dt} = S_v\left(\beta_h \frac{I_h}{N_h}\right) - (\mu_v + \gamma_v)I_v. \tag{34}$$

The parameter definitions are the same as for the model (1)–(5); however, they refer to hosts or vectors, depending on the subscript. The exceptions are the subscripts of $\beta$ and $\beta'$, which indicate that the disease transmissibility is from vectors to humans (i.e., subscript $v$) and humans to vectors (i.e., subscript $h$), respectively.

Neglecting the per-host natural-mortality rate, we obtain

$$R'_0 = \frac{\beta'_v \beta'_h}{(\mu_h + \gamma'_h + m'_h)(\mu_v + \gamma'_v)} \approx \frac{\beta'_v \beta'_h}{(\gamma'_h + m'_h)(\mu_v + \gamma'_v)} \tag{35}$$

for modeling the transmission of the wild type strain alone (i.e., the model consists of Eqs (27), (28), (30), (32) and (33), where $m'_h = 0$, $I_h = 0$ and $I_v = 0$), and

$$R_0 = \frac{\beta_v \beta_h}{(\mu_h + \gamma_h)(\mu_v + \gamma_v)} \approx \frac{\beta_v \beta_h}{\gamma_h(\mu_v + \gamma_v)} \tag{36}$$

for modeling the transmission of the mutant strain alone (i.e., the model consists of Eqs (27), (29), (31), (32) and (34), where $m'_h = 0$, $I'_h = 0$ and $I'_v = 0$). As before, we assume $R'_0 < 1$ and $R_0 > 1$.

We rewrite the model (27)–(34) at disease invasion, where $S_{h,v} \approx N_{h,v}$, and neglect the host demographic dynamics. We also introduce the parameter $N \equiv \pi_v \mu_h/(\pi_h \mu_v)$, estimating $N_v/N_h$, the number of vectors per capita, at the demographic equilibrium in absence of disease. We

therefore obtain

$$\frac{dI'_h}{dt} = (\beta'_v/\text{N})I'_v - \gamma'_h I'_h - m'_h I'_h, \tag{37}$$

$$\frac{dI_h}{dt} = m'_h I'_h + (\beta_v/\text{N})I_v - \gamma_h I_h, \tag{38}$$

$$\frac{dI'_v}{dt} = (\text{N}\beta'_h)I'_h - (\mu_v + \gamma'_v)I'_v, \tag{39}$$

$$\frac{dI_v}{dt} = (\text{N}\beta_h)I_h - (\mu_v + \gamma_v)I_v. \tag{40}$$

As before, a continuous-time Markov chain can be naturally defined, such that the biological processes described in Eqs (37)–(40) are represented as point processes; see Table 2. Each stochastic variable in the Markov chain corresponds to an ODE variable in Eqs (37)–(40). In fact, it can be checked straightforwardly that the moment expansion of the Markov chain variables closes at the expectation, and leads to Eqs (37)–(40), where

$$\langle i'_h \rangle = I'_h, \quad \langle i_h \rangle = I_h, \quad \langle i'_v \rangle = I'_v, \quad \langle i_v \rangle = I_v. \tag{41}$$

The extinction time of this Markov chain is defined as
$\inf\{t \geq 0 : i'_h(0) = 1, i_h(0) = 0, i'_v(0) = 0, i_v(0) = 0\}$, where the stochastic variables depend on the model parameters and initial condition. The extinction probability is given by the fraction of times the Markov chain trajectories go extinct. The extinction probability of the Markov chain represents the probability of emergence failure of the wild-type strain. However, if only the mutant strain circulates, then this represents the probability of epidemic extinction. Assuming that the epidemic started with $j$ patients zero and $k$ vectors zero, previous results [20, 79] yield

$$p_{jk} = \left(\frac{\gamma_h}{\beta_v/\text{N}}\right)^j \left(\frac{\mu_v + \gamma_v}{\text{N}\beta_h}\right)^k \left[\frac{\beta_v/\text{N} + (\gamma_v + \mu_v)}{\text{N}\beta_h + \gamma_h}\right]^{j-k}. \tag{42}$$

**Table 2. Markov chain for the model with vector-borne transmission: Stochastic processes and their corresponding rates.**

| Count $j$ | Process | Definition | Rate $\mathcal{R}_j$ |
|---|---|---|---|
| 1 | Host infection with the wild-type strain | $i'_h \to i'_h + 1$ | $(\beta'_v/\text{N})i'_v$ |
| 2 | Host recovery from infection with the wild-type strain | $i'_h \to i'_h - 1$ | $\gamma'_h i'_h$ |
| 3 | Mutation of the wild-type strain | $i'_h \to i'_h - 1, i_h \to i_h + 1$ | $m'_h i'_h$ |
| 4 | Host infection with the mutant strain | $i_h \to i_h + 1$ | $(\beta_v/\text{N})i_v$ |
| 5 | Host recovery from infection with the mutant strain | $i_h \to i_h - 1$ | $\gamma_h i_h$ |
| 6 | Vector infection with the wild-type strain | $i'_v \to i'_v + 1$ | $(\text{N}\beta'_h)i'_h$ |
| 7 | Vector natural mortality or mortality caused by infection with the wild-type strain | $i'_v \to i'_v - 1$ | $(\mu_v + \gamma'_v)i'_v$ |
| 8 | Vector infection with the mutant strain | $i_v \to i_v + 1$ | $(\text{N}\beta_h)i_h$ |
| 9 | Vector natural mortality or mortality caused by infection with the mutant strain | $i_v \to i_v - 1$ | $(\mu_v + \gamma_v)i_v$ |

We can write this as $p_{jk} = (p_{10})^j (p_{01})^k$. The factorization is intuitive since we model disease invasion, where the number of susceptibles is virtually infinite, and the transmission chains are independent. In the rest of this section, we will obtain an analytic formula for $p'_{jk}$, the probability of emergence failure for the wild-type strain.

We adapt ideas from the analysis of the model with direct transmission. We integrate the Markov chain in Table 2 for one time step, using Gillespie's algorithm. The initial condition is one patient minus one (i.e., $i'_h(0) = 1$, $i_h(0) = 0$, $i'_v(0) = 0$, $i_v(0) = 0$). Only Processes 2, 3, and 6 have non-zero rates at $t = 0$. The probabilities that they occur as the first process are, respectively, $\gamma'_h / (N\beta'_h + \gamma'_h + m'_h)$, $m'_h / (N\beta'_h + \gamma'_h + m'_h)$ and $N\beta'_h / (N\beta'_h + \gamma'_h + m'_h)$. We now formulate an equation for $p'_{10}$ based on the processes where patient minus one participates

$$p'_{10} = \frac{\gamma'_h}{N\beta'_h + \gamma'_h + m'_h} + \frac{p_{10} m'_h}{N\beta'_h + \gamma'_h + m'_h} + \frac{p'_{11}(N\beta'_h)}{N\beta'_h + \gamma'_h + m'_h}. \tag{43}$$

Eq (43) can be read as follows. Extinction of transmission starting from one patient minus one occurs because 1) the patient recovers with probability $\gamma'_h / (N\beta'_h + \gamma'_h + m'_h)$, or 2) the patient minus one becomes a patient zero with probability $m'_h / (N\beta'_h + \gamma'_h + m'_h)$, and forward transmission from the patient zero goes extinct with probability $p_{10}$, or 3) the patient minus one infects a vector, with probability $(N\beta'_h)/(N\beta'_h + \gamma'_h + m'_h)$, and forward transmission from the patient minus one and the vector minus one goes extinct with probability $p'_{11} = p'_{10} p'_{01}$. The unknowns in Eq (43) are $p'_{10}$ and $p'_{11} = p'_{10} p'_{01}$, so the equation is not sufficient.

We now integrate the Markov chain for one time step, where the initial condition is one vector minus one (i.e., $i'_h(0) = 0$, $i_h(0) = 0$, $i'_v(0) = 1$, $i_v(0) = 0$). In this case, only Processes 1 and 7 have non-zero rates at $t = 0$. They occur as the first process in the Markov chain with the respective probabilities $(\beta'_v/N)/(\beta'_v/N + \gamma'_v + \mu_v)$ and $(\gamma'_v + \mu_v)/(\beta'_v/N + \gamma'_v + \mu_v)$. We now obtain the following equation for $p'_{01}$

$$p'_{01} = \frac{p'_{11} \beta'_v/N}{\beta'_v/N + \gamma'_v + \mu_v} + \frac{\gamma'_v + \mu_v}{\beta'_v/N + \gamma'_v + \mu_v}. \tag{44}$$

The failure of transmission from one vector minus one is accounted as follows: the vector either 1) infects a host, with probability $(\beta'_v/N)/(\beta'_v/N + \gamma'_v + \mu_v)$, and forward transmission from the vector minus one and the patient minus one goes extinct with probability $p'_{11}$ or 2) dies, with probability $(\gamma'_v + \mu_v)/(\beta'_v/N + \gamma'_v + \mu_v)$.

Eqs (43) and (44) form a self-consistent system with two unknowns, which can be solved analytically. We multiply Eqs (43) and (44) to find a quadratic equation in $p'_{11}$;

$$p'_{11} = \left[ \frac{1 - (1 - p_{10})m'_h}{1 + R'_{0h}} + \frac{p'_{11} R'_{0h}}{1 + R'_{0h}} \right] \left( \frac{1}{1 + R'_{0v}} + \frac{p'_{11} R'_{0v}}{1 + R'_{0v}} \right), \tag{45}$$

where we changed notation to reduce the number of parameters ($R'_{0h} R'_{0v} = R'_0$) and have

$$R'_{0h} \equiv \frac{N\beta'_h}{\gamma'_h + m'_h}, \quad R'_{0v} \equiv \frac{\beta'_v/N}{\gamma'_v + \mu_v}, \quad m'_h \equiv \frac{m'_h}{\gamma'_h + m'_h}. \tag{46}$$

We discard the solution $p'_{11} = 1$, which only occurs when $p_{10} = 1$. In fact, since $R'_0 < 1$, Eq (45) has a unique solution to be interpreted as a probability

$$p'_{11} = \left[ \frac{1 + R'_0}{2R'_0} + \frac{m'_h(1 - p_{10})}{2R'_{0h}} \right] - \sqrt{\left[ \frac{1 + R'_0}{2R'_0} + \frac{m'_h(1 - p_{10})}{2R'_{0h}} \right]^2 - \frac{1 - m'_h(1 - p_{10})}{R'_0}}. \tag{47}$$

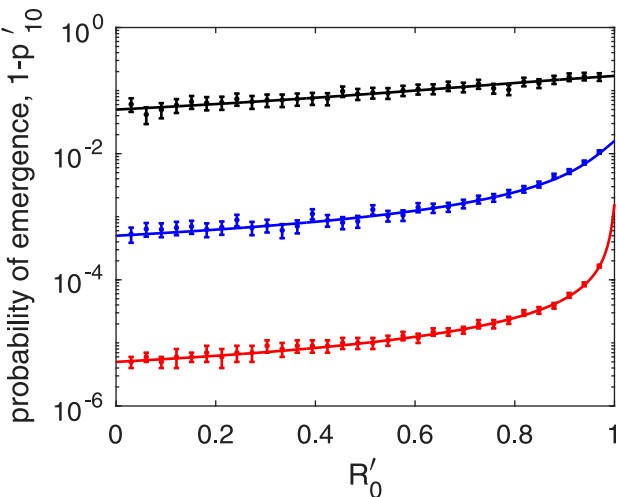

**Fig 5. The probability of emergence of vector-borne diseases.** We illustrate the probability that an epidemic emerges starting from just one patient minus one; i.e., $1 - p'_{10}$. The horizontal axis represents $R'_0$, the basic reproduction number associated to the infection of patient minus one. We chose three values for the mutation parameter, $\mathrm{m}'_h = 0.1$ (black plot), $\mathrm{m}'_h = 0.001$ (blue plot) and $\mathrm{m}'_h = 0.00001$ (red plot). The remaining parameters are $R'_{0h} = \sqrt{R'_0}$ and $p_{10} = 0.5$. The curves are obtained using Eq (50), and the points with error bars are obtained through brute-force integration of the Markov chain described in Table 2. Note that, with decreasing mutation rate, the probability curve develops a hyperbolic trend in the vicinity of $R'_0 = 1$.

Considering that the mutation rate $\mathrm{m}'_h$ is small, first-order calculations yield

$$p'_{11} \approx 1 - (1 + R'_{0v})\left[\frac{\mathrm{m}'_h(1 - p_{10})}{1 - R'_0}\right], \tag{48}$$

showing that, just like in the case of direct transmission, the key parameter driving the probability of emergence failure is the basic reproduction ratio of the wild type strain, $R'_0$. The probabilities $p'_{10}$ and $p'_{01}$ can be immediately obtained from substituting $p'_{11}$ in Eqs (43) and (44). For example, $p'_{10}$ can be written as

$$p'_{10}(R'_{0h}, \mathrm{m}'_h; p_{10}) = [(1 - \mathrm{m}'_h) + p_{10}\mathrm{m}'_h + p'_{11}R'_{0h}]/(R'_{0h} + 1). \tag{49}$$

In Fig 5, we illustrate the agreement between values of $p'_{10}$ provided by the above equation (continuous lines) and brute-force, repeated integration of the Markov chain in Table 2 (points with error bars). The integration of the Markov chain starts with the initial condition of one patient minus one (i.e., $i'_h(0) = 1$, $i_h(0) = 0$, $i'_v(0) = 0$, $i_v(0) = 0$) and terminates at time $t$ when $i'_h(t) + i_h(t) + i'_v(t) + i_v(t) = 0$ or $i_h(t) > 10,000$. The fraction of times that the population went extinct (i.e., $i'_h(t) + i_h(t) + i'_v(t) + i_v(t) = 0$) estimates the extinction probability of the Markov chain. The error bars estimate the 95% confidence interval of the probability of emergence.

As in Fig 2, the black plot refers to $\mathrm{m}'_h = 0.1$, the blue plot to $\mathrm{m}'_h = 0.001$, and the red plot to $\mathrm{m}'_h = 0.00001$. The horizontal axis represents $R'_0$, going from 0 to 1, and we chose $p_{10} = 0.5$. With this input, our choice of parameters for the Markov chain is: $\beta'_v = 1$, $\mathrm{N} = 1$, $\mathrm{m}'_h = \mathrm{m}'_h/\sqrt{R'_0}$, $\gamma'_h = 1/\sqrt{R'_0} - \mathrm{m}'_h$, $\beta_v = 1$, $\gamma_h = 1/3$, $\beta'_h = 1$, $\mu_v + \gamma'_v = 1/\sqrt{R'_0}$, $\beta_h = 1$, $\mu_v + \gamma_v = 1$. The number of repeat integrations are, respectively, 1000, 100,000, and 10,000,000. Overall, there is good agreement between the colored lines and the corresponding points with error bars.

The probability of emergence failure when the transmission process starts with $j$ patients minus one and $k$ vectors minus one is then given by $p'_{jk} = (p'_{10})^j (p'_{01})^k$. Finally, note that $\beta'_v = 0$ yields

$$p'_{10}(0, \mathrm{m}'_h; p_{10}) = 1 - \mathrm{m}'_h(1 - p_{10}), \tag{50}$$

which is similar to Eq (11). Thus, the probability of emergence, $(1 - p'_{10})$, equals the probability of acquiring the mutant strain $\mathrm{m}'_h$ times the probability that patient zero triggers an epidemic $(1 - p_{10})$.

## The case of $n$-step mutation

As in the case of direct transmission, we investigate the probability that pathogen emergence fails during an arbitrary long chain of neutral mutations. We assume that mutations occur only within hosts, yet the pathogen may be transmitted to, or be acquired from, vectors at any step in the mutation sequence, without consequences for the genetic dynamics.

The one-dimensional map that gives $p_{10}^{(k+1)}$ versus $p_{10}^{(k)}$ results from combining Eqs (47) and (49)

$$
\begin{aligned}
p_{10}^{(k+1)} = &\frac{1 - \mathrm{m}'_h(1 - p_{10}^{(k)})}{\mathrm{R}'_{0h} + 1} + \frac{\mathrm{R}'_{0h}}{\mathrm{R}'_{0h} + 1} \left\{ \left[ \frac{1 + \mathrm{R}'_0}{2\mathrm{R}'_0} + \frac{\mathrm{m}'_h(1 - p_{10}^{(k)})}{2\mathrm{R}'_{0h}} \right] \right. \\
&\left. - \sqrt{\left[ \frac{1 + \mathrm{R}'_0}{2\mathrm{R}'_0} + \frac{\mathrm{m}'_h(1 - p_{10}^{(k)})}{2\mathrm{R}'_{0h}} \right]^2 - \frac{1 - \mathrm{m}'_h(1 - p_{10}^{(k)})}{\mathrm{R}'_0}} \right\} \equiv f'_{\boldsymbol{\mu}}(p_{10}^{(k)}).
\end{aligned}
\tag{51}
$$

The map $f'_{\boldsymbol{\mu}}(\cdot)$ depends on a set of three parameters $\boldsymbol{\mu}' \equiv (\mathrm{R}'_{0h}, \mathrm{R}'_0, \mathrm{m}'_h)$ and has a fixed point at $\hat{p}_{10} = 1$, for all parameter values. If

$$\left. \frac{\partial f'_{\boldsymbol{\mu}}(p_{10}^{(k)})}{\partial p_{10}^{(k)}} \right|_{\hat{p}_{10}=1} = \frac{\mathrm{m}'_h}{(1 - \mathrm{R}'_0)} < 1, \tag{52}$$

then $\hat{p}_{10} = 1$ is stable and extinction is guaranteed. Fig 6 shows two graphs of $f'_{\boldsymbol{\mu}}(p_{10}^{(k)})$, for two different parameter sets, suggesting that $f'_{\boldsymbol{\mu}}(p_{10}^{(k)})$ undergoes a transcritical bifurcation. Redefining the parameter set

$$\boldsymbol{\mu}' = (\mathrm{R}'_{0h}, \mathrm{R}'_0, \mathrm{m}'_h) \rightarrow \left( \mathrm{R}'_{0h}, \mathrm{R}'_0, \lambda \equiv \frac{\mathrm{m}'_h}{(1 - \mathrm{R}'_0)} \right) \tag{53}$$

and choosing $\lambda$, the third parameter in the set, as the bifurcation parameter, the map $f'_{\boldsymbol{\mu}}(\cdot) \equiv f'_\lambda(\cdot)$ satisfies the conditions for having a transcritical bifurcation at $(\hat{p}^*_{10}, \lambda^*) = (1, 1)$ [78].

The existence of the transcritical bifurcation has important implications for the fixed-point dynamics. Increasing the mutation rate above the threshold $(1 - \mathrm{R}'_0)$ (i.e., $\mathrm{m}'_h > (1 - \mathrm{R}'_0)$ or $\lambda > 1$), $\hat{p}_{10} = 1$ becomes unstable and a second fixed point $\tilde{p}_{10} < 1$ becomes stable. The neutral mutation chain can survive extinction because the pathogen replicates and mutates fast enough, a mechanism similar to that identified for directly transmitted pathogens. In contrast, if the mutation rate is below the threshold $(1 - \mathrm{R}'_0)$ (i.e., $\mathrm{m}'_h < (1 - \mathrm{R}'_0)$ or $\lambda < 1$), then $\hat{p}_{10}$ is stable and extinction occurs with probability 1.

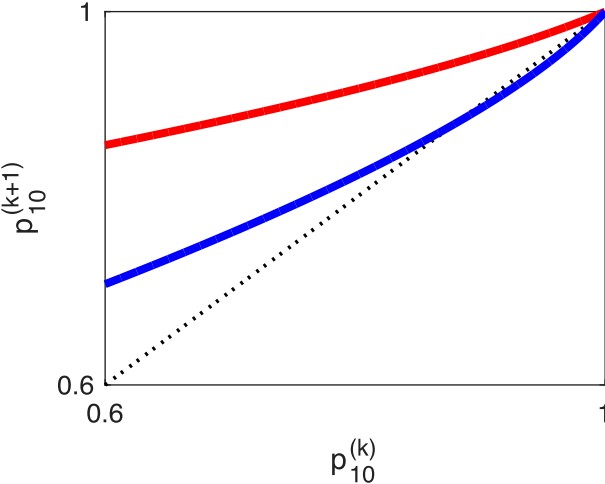

**Fig 6. Transcritical bifurcation of the neutral genetic dynamics for the pathogen with vector-borne transmission mechanism.** We illustrate $p_{10}^{(k+1)}$ versus $p_{10}^{(k)}$ or $f_{\mu}'(\cdot)$. The parameter sets are $(\mathrm{R}_{0h}', \mathrm{R}_0', \mathrm{m}_h') = (0.7, 0.7, 0.17)$ for the red curve and $(\mathrm{R}_{0h}', \mathrm{R}_0', \mathrm{m}_h') = (0.7, 0.7, 0.45)$ for the blue curve. The dotted line represents the first bisect line. It appears that $p_{10}^{(k+1)}(\cdot)$ undergoes a transcritical bifurcation at $(p_{10}^{(k)}, p_{10}^{(k+1)}) = (1, 1)$. That is, with the variation of the third parameter, the red curve becomes the blue curve, where $(1, 1)$ remains always a fixed point.

## Parameterization of models of pathogen emergence

The diverse and broad challenges in modeling the emergence of novel pathogens are discussed in Ref [82]. Here, we address the challenges in parameterizing Markov chain models using existing data. For the case of the zoonotic emergence of SARS-CoV1, the basic reproduction number of the wild-type pathogen $\mathrm{R}_0'$ [83, 84] and that of the emerged pathogen $\mathrm{R}_0$ [84] have been estimated, but not the mutation rate $\mathrm{m}'$. It is perhaps reasonable to assume that the mutations required for the emergence of SARS-CoV1 occurred just a few times, sufficient for the emerging SARS-CoV1 to spread worldwide. For a reliable estimate of the mutation rate $\mathrm{m}'$, many more mutation challenges should be documented, to reduce the uncertainty in the central estimate of $\mathrm{m}'$. In turn, this would allow for the estimation of $p_1'$, the probability of SARS-CoV1 to emerge. For other zoonoses such as MERS-CoV and monkeypox, epidemiological data is even more lacking. The subthreshold transmission of MERS-CoV [83, 84] and monkeypox [85] has been quantified. However, these zoonoses did not emerge as pandemics, so far, and data about the transmissibility of the emerged pathogen are missing. In the case of SARS-CoV2, the basic reproduction number of the wild-type pathogen in the human population was larger than 1 (i.e., $\mathrm{R}_0' > 1$) to begin with [86]. Therefore, SARS-CoV2 caused a pandemic among humans directly, without undergoing emergence.

Gathering all required parameters to model emergence of drug resistance seems difficult, but the difficulties are of different nature. It is broadly accepted that effective therapy can reduce the infectiousness of patients. Therefore, the transmission potential of a patient can decrease from $\mathrm{R}_0' > 1$ to $\mathrm{R}' < 1$; i.e, from the basic reproduction number of the running epidemic to the effective reproduction number of the epidemic of the treated disease. However, data to quantify this effect across infectious diseases is lacking. The only exception may be HIV, for which it has been documented that virally suppressed individuals undergoing HIV therapy no longer transmit HIV [87]; i.e., $\mathrm{R}' = 0$. Still, individuals may lose their state of viral suppression and, subsequently, develop virologic failure [88, 89]. Recent figures show that, in many settings, the rate of virologic failure is less than 10% [90–92], answering the 90-90-90

WHO/UNAIDS targets. Hence, virologic failure is considered a rare event in HIV therapy [88]. Furthermore, it is reasonable to assume that, with relentless efforts to optimize the HIV therapy, virologic failure will become even rarer [91, 93].

More than 70–80% of HIV patients developing virologic failure also develop drug-resistant HIV [89]. WHO surveys show that the prevalence of acquired drug resistance among people receiving ART in various settings, is 3–29% [94]; the total HIV mutation rate depends on the setting and decreases slowly with time. The database of the International Antiviral Society lists 171 clinically relevant, single mutations of HIV, associated with HIV drugs in five therapeutic classes [95]. However, typical drug-resistant HIV contains several single mutations, and the total number of relevant strains is much more than 171. Indeed, HIV has ample possibilities for evolution under drug pressure. To document HIV emergence, we have to stratify the total mutation rate by drug-resistant strain and identify the strains transmitted with $R > 1$. This requires much data, because HIV has many mutations relevant to treatment and the mutation rates are expected to be low. Instead, analyses often group drug-resistant HIV strains by resistance to the drug classes [88, 89, 96].

There is ample evidence that acquired drug-resistant HIV can be transmitted forward to others. In particular, the phrase *transmitted HIV drug resistance* refers to therapy-naïve individuals undergoing primary infection due to drug resistant HIV. The prevalence of transmitted drug resistance by standard genotypic resistance testing is about 12–24% in the U.S., 5–10% in Europe, Latin America, and high-income Asian countries, and less than 5% for most of the sub-Saharan Africa and South/Southeast Asia [89]. In principle, an individual harboring drug-resistant HIV, not being virally suppressed because of this, can transmit HIV in two different circumstances. First, the individual undergoes HIV therapy and thus the drug-resistant strains have a higher within-host fitness than the fully-susceptible strains, which remain suppressed. The individual can pass drug resistant HIV to others, unless the virological failure is detected and resistance tests are performed. Subsequently, changing the HIV regimen to the next line will likely bring the individual back to viral suppression, so he no longer transmits HIV. Secondly, the individual does not undergo HIV therapy. In this case, the drug-resistant strains have a lower within-host fitness than the fully susceptible strains. It is thus only matter of time—typically, several years—that HIV gradually loses its drug-resistant mutations and reverts to wild type [89]. Hence, the individual can pass drug-resistant HIV to others for only a window of time.

The epidemiological dynamics of HIV drug resistance have often been modeled mathematically. However, only on rare occasions drug resistance has been stratified [97, 98], for a glimpse into how the competition for emergence of drug-resistant HIV takes place. To the best of our knowledge, drug-resistance dynamics have never been fully stratified by mutation strain in a mathematical model of the HIV epidemic. Rather, they have been stratified by drug-resistance class, and a spectrum of effective reproduction numbers has been calculated as a function of the resistance class [97]. However, the notion of strain representative for an entire drug class may be ill-defined and find no justification in the biology of pathogen emergence. The proper modeling exercise would need to stratify the epidemic dynamics over all relevant mutations. In the case of HIV, this yields a very large epidemic model, which is very difficult to parameterize.

## Discussion

Understanding the mechanisms by which pathogens emerge to establish widespread epidemics remains a fundamental question in epidemiology. Broadly speaking, we distinguish zoonotic emergence and emergence of drug resistance. EIDs can be initiated by poorly adapted

pathogens, whose transmission takes places with $R_0'$ (or $R'$) less than 1. However, the pathogens can break the evolutionary barrier and mutate to be transmitted with $R_0$ (or $R$) larger than 1 and eventually lead to endemic disease; e.g., SARS-CoV1 and HIV.

We called an individual infected by the poorly adapted pathogen, patient minus one. We obtained analytic formulae for the probability that patient minus one fails to initiate an epidemic of a directly transmitted or vector-borne disease. Two configurations for the mutation landscape were explored, where the original pathogen is 1) one step-mutation away from the epidemic strain, and 2) undergoing a long chain of neutral mutations that do not change the epidemic-level parameters.

We obtained analytic results for the probabilities of emergence failure. Illustrations of these analytic and numerical results (i.e., Figs 2 and 5) appear similar to the numerical results by Antia et al. [24] (i.e., Fig 2a). Furthermore, we obtained two features that remain valid for both transmission mechanisms. First, as noted by Antia et al. [24], the reproduction number of the original pathogen is determinant for the probability of pathogen emergence, more important than the mutation rate or the transmissibility of the emerged pathogen. Secondly, the probability of mutation within infected individuals must be sufficiently high for the pathogen undergoing neutral mutation to start an epidemic. The threshold depends on the basic reproduction number of the original pathogen; i.e., $m'$ or $m_h'$ must be larger than $(1 - R_0')$ for zoonotic emergence and $(1 - R')$ for the emergence of drug resistance. Each particular strain has reproduction number less than 1 and is guaranteed to go extinct. However, the strain mix can persist and be transmitted forward. The key dynamical features are transcritical bifurcations, illustrated in Figs 3 and 6. Finally, we argued that parameterizing models of pathogen emergence to represent the epidemiology of SARS-CoV1 or HIV is a difficult endeavor.

A key assumption, inherited from previous work [19, 20], was that of disease invasion, where infectious individuals undergo proportional mixing with an arbitrarily large number of susceptibles. The assumption holds well for the early stages of epidemics when depletion of susceptibles is negligible. The resulting mathematical simplification cannot be overstated. It makes the transmission chains independent, and depletion of susceptibles no longer occurs, for all times. It also implies that competing epidemics at disease invasion are independent.

As criticism for this assumption, Kubiak et al. [57] noted that many zoonotic cases, originating from wild and/or domestic animals, occur in small communities, away from large population centers. Emergence through drug resistance may also occur within a similar community structure. A smaller community with less than perfect drug management can put a larger community at risk. Hence, EIDs often occur in heterogeneous communities. An interesting modeling account of EIDs in heterogeneous host populations, based on branching processes, is given by Chabas et al. [99].

Communities have finite populations and do not escape the phenomenon of depletion of susceptibles, except for a short while, at the very beginning of epidemics. Emergence failure in heterogeneous, small communities is more likely than in communities with very large number of susceptibles. Therefore, our formulae for the probability of emergence failure provide upper bounds for more realistic results describing heterogeneous communities of finite size. In fact, it is important to note that the Markov chain extinction problem is soluble numerically. Markov chains of arbitrary complexity can be integrated numerically for a computational surrogate of the probability of emergence failure. Furthermore, parameter hierarchies can also be established numerically through uncertainty and sensitivity analyses of the probability result. There is no need, at any point in the analysis, to invoke a branching process.

## Conclusions

Markov chains are the modeling tools of choice to describe disease invasion analytically. It is only natural to generalize these models to include the principles of disease emergence. The required mathematics remains simple enough that analytic solutions for the extinction probability can still be obtained. If the assumption of disease invasion does not hold, then Markov chains still can be integrated numerically to estimate the probability of emergence. Branching processes do not typically offer this convenience. The popular branching-process model implicitly assumes the setup of disease invasion. However, the major difficulty in modeling disease emergence remains model parameterization, based on current epidemiological data.

## Acknowledgments

The author thanks Virginie Supervie for reading the manuscript and discussions. The author also thanks two referees for constructive criticism.

## Author Contributions

**Conceptualization:** Romulus Breban.

**Formal analysis:** Romulus Breban.

**Methodology:** Romulus Breban.

**Software:** Romulus Breban.

**Validation:** Romulus Breban.

**Writing – original draft:** Romulus Breban.

**Writing – review & editing:** Romulus Breban.

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
