## [Decision Letter · Decision Letter 0]

27 Sep 2023

PONE-D-23-17372Emergence failure of early epidemics: A mathematical modeling approachPLOS ONE

Dear Dr. Breban,

Thank you for submitting your manuscript to PLOS ONE. After careful consideration, we feel that it has merit but does not fully meet PLOS ONE’s publication criteria as it currently stands. Therefore, we invite you to submit a revised version of the manuscript that addresses the points raised during the review process.

We look forward to receiving your revised manuscript.

Kind regards,

Kristan Alexander Schneider

Academic Editor

PLOS ONE

Journal Requirements:

Reviewers' comments:

Reviewer's Responses to Questions

**Comments to the Author**

1. Is the manuscript technically sound, and do the data support the conclusions?

Reviewer #1: Yes

Reviewer #2: Yes

2. Has the statistical analysis been performed appropriately and rigorously? 

Reviewer #1: N/A

Reviewer #2: N/A

3. Have the authors made all data underlying the findings in their manuscript fully available?

Reviewer #1: Yes

Reviewer #2: Yes

4. Is the manuscript presented in an intelligible fashion and written in standard English?

Reviewer #1: Yes

Reviewer #2: No

5. Review Comments to the Author

Reviewer #1: Summarizing, I found the paper interesting and well-written. Nevertheless, as probabilistic models are not my strong point, I would appreciate if the link with the dynamics of the original deterministic models was better explained.

See the attached file.

Reviewer #2: See attached marked-up manuscript.

See attached marked-up manuscript.

See attached marked-up manuscript.

See attached marked-up manuscript.

See attached marked-up manuscript.

(The minimum character count is ridiculous when there are attachments.)

6. PLOS authors have the option to publish the peer review history of their article (what does this mean?). If published, this will include your full peer review and any attached files.

Reviewer #1: No

Reviewer #2: No

---

## [Author Response · Author response to Decision Letter 0]

9 Nov 2023

Please see attached Response to Reviewers.

---

## [Decision Letter · Decision Letter 1]

29 Jan 2024

PONE-D-23-17372R1Emergence failure of early epidemics: A mathematical modeling approachPLOS ONE

Dear Dr. Breban,

Thank you for submitting your manuscript to PLOS ONE. After careful consideration, we feel that it has merit but does not fully meet PLOS ONE’s publication criteria as it currently stands. Therefore, we invite you to submit a revised version of the manuscript that addresses the points raised during the review process.

 Reviewer #2 made some edits that should be incorporated in the revised version of the manuscript. 

We look forward to receiving your revised manuscript.

Kind regards,

Kristan Alexander Schneider

Academic Editor

PLOS ONE

Journal Requirements:

Reviewers' comments:

Reviewer's Responses to Questions

**Comments to the Author**

1. If the authors have adequately addressed your comments raised in a previous round of review and you feel that this manuscript is now acceptable for publication, you may indicate that here to bypass the “Comments to the Author” section, enter your conflict of interest statement in the “Confidential to Editor” section, and submit your "Accept" recommendation.

Reviewer #1: All comments have been addressed

Reviewer #2: (No Response)

2. Is the manuscript technically sound, and do the data support the conclusions?

Reviewer #1: Yes

Reviewer #2: Yes

3. Has the statistical analysis been performed appropriately and rigorously? 

Reviewer #1: Yes

Reviewer #2: Yes

4. Have the authors made all data underlying the findings in their manuscript fully available?

Reviewer #1: Yes

Reviewer #2: Yes

5. Is the manuscript presented in an intelligible fashion and written in standard English?

Reviewer #1: Yes

Reviewer #2: Yes

6. Review Comments to the Author

Reviewer #1: My all concerns have been appropriately addressed and the paper can be published in its current form.

Reviewer #2: See attached marked-up manuscript.

Please use the space provided to explain your answers to the questions above. You may also include additional comments for the author, including concerns about dual publication, research ethics, or publication ethics. (Please upload your review as an attachment if it exceeds 20,000 characters) (Limit 100 to 20000 Characters) <-- Why is there a minimum when you can upload an attachment???

7. PLOS authors have the option to publish the peer review history of their article (what does this mean?). If published, this will include your full peer review and any attached files.

Reviewer #1: No

Reviewer #2: **Yes: **Stacey Smith?

---

## [Editor Report · Decision Letter 2]

17 Mar 2024

Emergence failure of early epidemics: A mathematical modeling approach

PONE-D-23-17372R2

Dear Dr. Breban,

We’re pleased to inform you that your manuscript has been judged scientifically suitable for publication and will be formally accepted for publication once it meets all outstanding technical requirements.

Kind regards,

Kristan Alexander Schneider

Academic Editor

PLOS ONE

Additional Editor Comments (optional):

The reviewers comments were appropriately addressed.
---

## [Editor Report · Acceptance letter]

2 May 2024

PONE-D-23-17372R2 

PLOS ONE

Dear Dr. Breban, 

I'm pleased to inform you that your manuscript has been deemed suitable for publication in PLOS ONE. Congratulations! Your manuscript is now being handed over to our production team.

Kind regards, 

on behalf of

Professor Kristan Alexander Schneider 

Academic Editor

PLOS ONE